# A New Iterative Algorithm for Magnetic Motion Tracking

**DOI:** 10.3390/s24216947

**Published:** 2024-10-29

**Authors:** Tobias Schmidt, Johannes Hoffmann, Moritz Boueke, Robert Bergholz, Ludger Klinkenbusch, Gerhard Schmidt

**Affiliations:** 1Digital Signal Processing and System Theory, Institute of Electrical Engineering and Information Technology, Faculty of Engineering, Kiel University, Kaiserstr. 2, 24143 Kiel, Germany; tsc@tf.uni-kiel.de (T.S.); jph@tf.uni-kiel.de (J.H.); mobo@tf.uni-kiel.de (M.B.); 2Computational Electromagnetics, Institute of Electrical Engineering and Information Technology, Faculty of Engineering, Kiel University, Kaiserstr. 2, 24143 Kiel, Germany; lbk@tf.uni-kiel.de; 3Pediatric Surgery, University Hospital Schleswig-Holstein, Kiel University, Arnold-Heller Str. 3, 24105 Kiel, Germany; robert.bergholz@uksh.de

**Keywords:** magnetic motion tracking, localization, rotating magnetic dipole, iterative algorithms, human–machine interface

## Abstract

Motion analysis is of great interest to a variety of applications, such as virtual and augmented reality and medical diagnostics. Hand movement tracking systems, in particular, are used as a human–machine interface. In most cases, these systems are based on optical or acceleration/angular speed sensors. These technologies are already well researched and used in commercial systems. In special applications, it can be advantageous to use magnetic sensors to supplement an existing system or even replace the existing sensors. The core of a motion tracking system is a localization unit. The relatively complex localization algorithms present a problem in magnetic systems, leading to a relatively large computational complexity. In this paper, a new approach for pose estimation of a kinematic chain is presented. The new algorithm is based on spatially rotating magnetic dipole sources. A spatial feature is extracted from the sensor signal, the dipole direction in which the maximum magnitude value is detected at the sensor. This is introduced as the “maximum vector”. A relationship between this feature, the location vector (pointing from the magnetic source to the sensor position) and the sensor orientation is derived and subsequently exploited. By modelling the hand as a kinematic chain, the posture of the chain can be described in two ways: the knowledge about the magnetic correlations and the structure of the kinematic chain. Both are bundled in an iterative algorithm with very low complexity. The algorithm was implemented in a real-time framework and evaluated in a simulation and first laboratory tests. In tests without movement, it could be shown that there was no significant deviation between the simulated and estimated poses. In tests with periodic movements, an error in the range of 1° was found. Of particular interest here is the required computing power. This was evaluated in terms of the required computing operations and the required computing time. Initial analyses have shown that a computing time of 3 μs per joint is required on a personal computer. Lastly, the first laboratory tests basically prove the functionality of the proposed methodology.

## 1. Introduction

Human motion tracking is of great interest to many applications such as virtual/ augmented reality [1] and medical diagnostics [2]. Among the several variants of motion tracking, this contribution focuses on hand-motion tracking as a human–machine interface for robot-assisted surgery. However, the proposed method can also be used for other applications where the movement can be modelled by kinematic chains.

Camera-based (optical motion capture, OMC) systems, which are considered to be the gold standard in motion tracking methods, allow for accuracy in the millimeter or even submillimeter range [3]. However, OMC systems have the disadvantage that direct lines of sight between the objects (usually reflecting markers) and relevant cameras are required.

An alternative method is the use of gloves with attached inertial measurement units (IMU) or flex sensors. Several of the corresponding solutions are shown in [4]. The well-investigated IMU are used in commercial applications such as *XSens’ Quantum Metaglove* [5]. However, IMU-based systems do not measure the quantities of interest (i.e., positions and angles) directly but instead measure their time derivatives (i.e., acceleration and angular speed). This leads to drift problems.

Approaches based on magnetic sensors are still in the early phases of research (see [6,7], for examples). Future magnetic methods could be either a stand-alone functional alternative or be used as a supplement to improve the performance of optical or IMU-based systems.

The present work aims to design an input device for robot-assisted surgery based on magnetic sensor technology. For this purpose, a glove will be equipped with magnetic sensors such that each kinematic element can be assigned to at least one sensor. The heart of the proposed motion tracking system is the localization unit. This unit determines the pose of the object with a constant sample rate with a typical duration of the period of 15 ms, which in turn leads to a localization update rate of 67 Hz [8,9]. If the sampling period is longer, it might lead to disruptive handling in human–machine interface applications. In our case, a kinematic chain with up to 20 degrees of freedom has to be estimated every 15 ms. The allowed latency is one of the challenges, as it leads to limited computing time and the algorithms must be designed to work efficiently.

Magnetic localization is usually solved with numerical or analytical approaches. Numerical solutions are used in applications with 1D sensors or flexible setups [10,11,12,13]. Analytical methods are used with fixed sensor array configurations, such as 3D sensors [14,15,16] or gradient sensors [17,18,19,20]. On the one hand, numerical methods are generally computationally intensive, which can become a problem if many (>20) sensors are involved, as it may no longer be possible to maintain the latency time. On the other hand, analytical methods usually use defined sensor setups such as 3D sensors. These may already be too large for the structures to be observed, which in case of a finger are only a few centimetres in size. In [6], a magnetic sensor glove with a numerical localization approach is described.

In the approaches mentioned above, while poses are determined by a minimization of a cost function and some kinematic constraints are kept, up to 55 hand reconstructions per second had been achieved thus far. Since about at least 67 hand reconstructions per second are required for surgical interfaces, these algorithms are not fully capable of solving the problem at this time. With the progress in computer hardware, these algorithms could become able to satisfy these conditions in the future. However, we are looking for a solution that can be executed on standard personal computers where further processing beyond motion analysis (e.g., gesture recognition) usually need to be executed.

For the overall system, we use two nested localization methods: The *external* localization is responsible for the absolute localization of a reference point within a defined measurement volume. In our case, this reference point could be the wrist, for example. A 3D coil is attached to this wrist, which can then be localized using conventional algorithms, as shown in [15,16]. Based on this, there is an *internal* localization. This estimates the position and orientation of the sensors attached to the fingers with respect to the reference point mentioned above. To this end, the individual fingers are modelled as kinematic chains. This offers an advantage in that the number of degrees of freedom is reduced. In general, a 1D sensor has five degrees of freedom: three for the position and two for the orientation. By attaching the sensor to the kinematic chain, the number of degrees of freedom per sensor is reduced to two. Here, movement is limited by the rotation of the joints. This will be utilized in an efficient algorithm, which will be explained in the following. An advantage of the presented algorithm is that it combines localization and mapping to a kinematic chain. In this way, prior knowledge about a kinematic chain is integrated into a localization, thus narrowing down the solution space and simplifying the calculation. An overview of this nested localization scheme is shown in Figure 1.

First, we will introduce a magnetic signal feature called the “maximum vector”. Then, we discuss the relations between this feature, the sensor position, and the sensor orientation. Eventually, these relations are linked to the known anatomy through an iterative algorithm. For validation, the algorithm was implemented in a real-time environment and tested with a simulation and an initial laboratory setup. The paper closes with a discussion about the results and the restrictions of the algorithm. The central idea and a suitable setup is shown in Figure 2.

## 2. Spatial Signal Feature

In this section, a magnetic signal feature will be introduced, which we refer to as the “maximum vector”, abbreviated as MV in the following. A rotating magnetic dipole at a defined position and a sensor at a defined position and orientation are assumed. The MV describes the dipole orientation for which the maximum signal is detected at the sensor. The spatial relationships between the MV, the sensor location, and the sensor orientation will be derived in the following.

### 2.1. Field-Theoretical Basics

The magnetic field of a slowly time-varying current density distribution can be approximated by Biot–Savart’s law. However, even this simplification of Maxwell’s equations can result in a very time-consuming numerical calculation, requiring a spatial integration over each voxel of the current–density distribution. For a localized current–density distribution produced by coils, the magnetic field calculation can be further simplified using a magnetic dipole model. For a good approximation, it is important that the distance between the coil and the sensor is much larger than the typical dimensions of the coil. For a magnetic dipole $\vec{m}$ located at the origin (r=0), the magnetic field ${\vec{B}}_{\mathrm{dip}}\left(\vec{r}\;\right)$ can described by [21]:(1)$${\vec{B}}_{\mathrm{dip}}\left(\vec{r}\right)=\frac{1}{4\pi r^{2}}\frac{3\vec{r}\left(\vec{m}\;\cdot \;\vec{r}\right)-\vec{m}r^{2}}{r^{3}}.$$

In the present work, we assume an ideal sensor at position $\vec{r}$ with known orientation ${\vec{e}}_{s}$. “Ideal” means that the sensor is assumed to be located at a single point and that the output of the sensor is an undisturbed projection of the dipole field on the main sensor axis. The output $B_{\mathrm{sensor}}\left(\vec{r},{\vec{e}}_{s}\right)$ is then found as follows:(2)$$B_{\mathrm{sensor}}\left(\vec{r},{\vec{e}}_{s}\right)={\vec{B}}_{\mathrm{dip}}\left(\vec{r}\right)\;\cdot \;{\vec{e}}_{s}.$$

For the generation of the magnetic field ${\vec{B}}_{\mathrm{dip}}$, we apply the 3D coil as represented in Figure 3. It consists of a superposition of three orthogonal coils. For the current application, this source can be represented by a superposition of three magnetic dipoles polarized in the *x*-, *y*-, and *z*-directions, respectively. Clearly, by appropriately weighting the amplitudes (i.e., currents) of the orthogonal three coils, an arbitrary single dipole can be created. Eventually, this will be used to form a single dipole that spatially rotates as a function of time. Figure 2 shows the setup. The source is located at the origin and the sensors are aligned with the chain elements.

### 2.2. Maximum Vector (MV)

We assume a sensor at an arbitrary position $\vec{r}$ with an arbitrary orientation ${\vec{e}}_{s}$ where both values are unknown. A rotating dipole source is located at the origin of the global coordinate system r=0. We define MV ${\vec{e}}_{\max}$ as the orientation of the dipole field ${\vec{B}}_{\mathrm{dip}}$ for which the sensor signal $B_{\mathrm{sensor}}$ in Equation (Equation 2) becomes a maximum.

We first derive a relationship between the direction of the sensor position ${\vec{e}}_{r}$, the sensor orientation ${\vec{e}}_{s}$, and the MV ${\vec{e}}_{\max}$. To this end, we first normalize the Equation (Equation 1) according to
(3)$${\vec{B}}_{\mathrm{norm}}=\frac{4\pi r^{3}}{m}{\vec{B}}_{\mathrm{dip}}\left(\vec{r}\right)=3{\vec{e}}_{r}\left({\vec{e}}_{m}\;\cdot \;{\vec{e}}_{r}\right)-{\vec{e}}_{m},$$
where $\vec{m}=m{\vec{e}}_{m}$.

Without loss of generality, we next assume that ${\vec{e}}_{r}$ and ${\vec{e}}_{s}$ lie in the xy-plane. As sketched in Figure 4, for the Cartesian coordinates of ${\vec{e}}_{r}$ and ${\vec{e}}_{s}$ we have
(4)$${\vec{e}}_{r}=\left[\begin{array}{c} \cos(\phi )\\ \sin(\phi )\\ 0 \end{array}\right]\;\mathrm{and}\;{\vec{e}}_{s}=\left[\begin{array}{c} \cos(\phi_{s})\\ \sin(\phi_{s})\\ 0 \end{array}\right].$$

The magnetic dipole moment $\vec{m}$ can point in any direction. In spherical coordinates, the Cartesian components of ${\vec{e}}_{m}$ read (see Figure 4)
(5)$${\vec{e}}_{m}=\left[\begin{array}{c} \cos\left(\phi_{m}\right)\sin\left(\theta_{m}\right)\\ \sin\left(\phi_{m}\right)\sin\left(\theta_{m}\right)\\ \cos(\theta_{m}) \end{array}\right].$$

Inserting Equations (Equation 4) and (Equation 5) into Equation (Equation 3) yields the three Cartesian field components of the normalized dipole field:
(6)$$\begin{array}{ccc} B_{\mathrm{norm},x}\left(\phi ,\phi_{m},\theta_{m}\right) & = & \left[3\cos{\left(\phi \right)}^{2}-1\right]\;\cos\left(\phi_{m}\right)\;\sin\left(\theta_{m}\right)\\ & + & 3\;\sin\left(\phi_{m}\right)\;\sin\left(\theta_{m}\right)\;\sin\left(\phi \right)\;\cos\left(\phi \right), \end{array}$$(7)$$\begin{array}{ccc} B_{\mathrm{norm},y}\left(\phi ,\phi_{m},\theta_{m}\right) & = & \left[3\sin{\left(\phi \right)}^{2}-1\right]\;\sin\left(\phi_{m}\right)\;\sin\left(\theta_{m}\right)\\ & + & 3\;\cos\left(\phi_{m}\right)\;\sin\left(\theta_{m}\right)\;\cos\left(\phi \right)\;\sin\left(\phi \right), \end{array}$$(8)$$\begin{array}{ccc} B_{\mathrm{norm},z}\left(\theta_{m}\right) & = & -\cos(\theta_{m}). \end{array}$$

The sensor signal is obtained using Equations (Equation 2)–(Equation 4) according to
(9)$$B_{\mathrm{sensor}}\left(\phi ,\phi_{s},\phi_{m},\theta_{m},r\right)=\frac{m}{4\pi r^{3}}\left((\cos\left(\phi_{s}\right)\;B_{\mathrm{norm},x}\left(\phi ,\phi_{m},\theta_{m}\right)+\sin\left(\phi_{s}\right)\;B_{\mathrm{norm},y}\left(\phi ,\phi_{m},\theta_{m}\right)\right).$$

Obviously, the variation in the distance between the sensor and the rotating dipole affects the measured signal according to $\frac{1}{r^{3}}$. Regarding the variation in $\theta_{m}$, the measured signal becomes a maximum only if $\theta_{m}=\pi /2$. Moreover, we remark that the component of the rotating dipole which is perpendicular to the plane spanned by ${\vec{e}}_{r}$ and ${\vec{e}}_{s}$ has no effect on the measured signal. From these observations we deduce that the MV must also lie in that plane. To find the desired relationship between the three unit vectors, we—without limiting the generality—place the location vector $\vec{r}$ on the *x*-axis (see the left side of Figure 5). For the Cartesian components of the corresponding unit vectors we thus have:(10)$${\vec{e}}_{r}=\left[\begin{array}{c} 1\\ 0 \end{array}\right],\;{\vec{e}}_{m}=\left[\begin{array}{c} \cos(\phi_{m})\\ \sin(\phi_{m}) \end{array}\right],\;{\vec{e}}_{s}=\left[\begin{array}{c} \cos(\phi_{s})\\ \sin(\phi_{s}) \end{array}\right].$$

In that case, it holds
(11)$$\begin{array}{ccc} B_{\mathrm{norm},x}\left(\phi \right) & = & 2\cos(\phi_{m}), \end{array}$$
(12)$$\begin{array}{ccc} B_{\mathrm{norm},y}\left(\phi \right) & = & -\sin(\phi_{m}), \end{array}$$
(13)$$\begin{array}{ccc} B_{\mathrm{norm},z}\left(\phi \right) & = & 0, \end{array}$$
and Equation (Equation 3) simplifies to
(14)$$B_{\mathrm{sensor}}\left(\phi_{s},\phi_{m}\right)=\frac{m}{4\pi r^{3}}\left(2\cos\left(\phi_{m}\right)\cos\left(\phi_{s}\right)-\sin\left(\phi_{m}\right)\sin\left(\phi_{s}\right)\right).$$

To derive the angle $\phi_{m}=\phi_{\max}$ where the right-hand side of Equation (Equation 14) becomes a maximum, we calculate
(15)$$\frac{dB_{\mathrm{sensor}}\left(\phi_{m}\right)}{d\phi_{m}}=\frac{m}{4\pi r^{3}}\left(-2\sin\left(\phi_{m}\right)\cos\left(\phi_{s}\right)-\sin\left(\phi_{s}\right)\cos\left(\phi_{m}\right)\right)$$
(16)$$-2\sin\left(\phi_{\max}\right)\cos\left(\phi_{s}\right)-\sin\left(\phi_{s}\right)\cos\left(\phi_{\max}\right)=0,$$
and finally obtain the relation:(17)$$-2\cdot \tan\left(\phi_{\max}\right)=\tan\left(\phi_{s}\right).$$
Note that this relation is valid for any ϕ (not just for ϕ=0), as sketched in Figure 5.

As graphically demonstrated in Figure 5 there is a unique relation between the three unit vectors ${\vec{e}}_{s},{\vec{e}}_{max},$ and ${\vec{e}}_{r}$. This relation will now be used to uniquely derive the sensor orientation ${\vec{e}}_{s}$ from a given ${\vec{e}}_{r}$ and a measured MV ${\vec{e}}_{\max}$. As an example, for a systematic procedure we start from a given origin (i.e., the location of the “rotating” 3D coil) and a given sensor location $\vec{r}$.

First, we determine the “rotation axis” ${\vec{e}}_{n}$:
(18)$${\vec{e}}_{n}=\frac{{\vec{e}}_{\max}\times {\vec{e}}_{r}}{∥{\vec{e}}_{\max}\times {\vec{e}}_{r}∥}.$$In the second step, the angle between the MV and the location unit vector is calculated:
(19)$$\phi_{\max}=\arccos\left({\vec{e}}_{\max}\;\cdot \;{\vec{e}}_{r}\right).$$Subsequently, the angle between the location unit vector and the sensor orientation is determined from Equation (Equation 17):
(20)$$\phi_{s}=\arctan\left(-2\tan(\phi_{\max})\right).$$Finally, for any given ${\vec{e}}_{r}$, the sensor orientation ${\vec{e}}_{s}$ is calculated by
(21)$${\vec{e}}_{s}=\cos\left(\phi_{s}\right)\left({\vec{e}}_{n}\times {\vec{e}}_{r}\right)\times {\vec{e}}_{n}+\sin\left(\phi_{s}\right)\left({\vec{e}}_{n}\times {\vec{e}}_{r}\right).$$

Figure 6 shows the calculated sensor orientations as blue vectors starting at different sensor locations $\vec{r}$ in the xy-plane. The MV is located at the origin and polarized in the *y*-direction. For the 3D case, i.e., if $\vec{r}$ is an arbitrary vector, we simply have to rotate the blue sensor orientations around the MV.

## 3. Signal Processing

In this section, it is shown how (temporal) signal processing can be used to extract the required signal feature. Subsequently, an iterative algorithm for pose estimation is presented.

### 3.1. Feature Extraction

The three orthogonal coils are driven with three different current signals. These currents are chosen such that the absolute value of the dipole moment $|\vec{m}\left(t\right)|=m_{0}$ is constant for all values of *t*:(22)$$\vec{m}\left(t\right)=m_{0}\left[\begin{array}{c} \cos\left(\omega_{\phi }t\right)\sin\left(\omega_{\theta }t\right)\\ \sin\left(\omega_{\phi }t\right)\sin\left(\omega_{\theta }t\right)\\ \cos(\omega_{\theta }t) \end{array}\right].$$

In particular, for the circle frequencies it holds that $\omega_{\theta }=N_{\omega }\omega_{\phi }$, where $N_{\omega }$ is a positive integer. Thus, the *x* and *y* coils are driven by an amplitude-modulated signal while the *z* coil produces a standard harmonic magnetic signal. Consequently, as a function of time, $\vec{m}$ moves on the surface of a sphere with radius $m_{0}$, as exemplary shown for $N_{\omega }=10$ in Figure 7.

While searching for the MV, an obvious method would be to try each direction to detect the maximum field within a given period of time. Instead, we will prove in the following that the directions where no field is measured are orthogonal to the MV. We call such vectors zero-crossing vectors. By detecting two independent zero-crossing vectors, the MV can then be calculated by simply building their $\phi_{\max}$, we first set Equation (Equation 14) (which is valid if ${\vec{e}}_{m}$ and ${\vec{e}}_{s}$ are lying in the xy-plane) for $\phi_{m}=\phi_{\mathrm{zero}}$ to zero: (23)$$\begin{array}{ccc} 0 & = & 2\;\cos\left(\phi_{\mathrm{zero}}\right)\;\cos\left(\phi_{s}\right)-\sin\left(\phi_{\mathrm{zero}}\right)\;\sin\left(\phi_{s}\right) \end{array}$$(24)$$\begin{array}{ccc} 2\cot(\phi_{\mathrm{zero}}) & = & \tan(\phi_{s}). \end{array}$$

With Equation (Equation 17) we have
(25)$$\begin{array}{ccc} \cot\left(\phi_{\mathrm{zero}}\right) & = & \tan\left(-\phi_{\max}\right) \end{array}$$
(26)$$\begin{array}{ccc} \cot\left(\phi_{\mathrm{zero}}\right) & = & \cot\left(\frac{\pi }{2}+\phi_{\max}\right) \end{array}$$
(27)$$\begin{array}{ccc} \phi_{\mathrm{zero}} & = & \phi_{\max}\pm \frac{\pi }{2}. \end{array}$$

Equation (27) proves that the zero-crossing vector and the MV are orthogonal if both are lying in the xy-plane. Since zero-crossing vectors and MVs are orthogonal, we conclude that for the general case the zero-crossing vectors can again be obtained by a rotation around the MV. Figure 8 illustrates an arbitrarily directed MV, the corresponding plane of zero-crossing vectors, and the direction of the sensor.

### 3.2. Zero-Crossing Polarity

In the previous section, we showed that the MV is orthogonal to all zero-crossing vectors. Using the cross product, a vector can be determined that is perpendicular to them. There are two solutions to this condition. For example, in the xy-plane the normal vectors are ${\vec{e}}_{z}$ and $-{\vec{e}}_{z}$. Therefore, it is necessary to observe the polarity here. Each zero-crossing is assigned a polarity that depends on two parameters. The direction of the zero-crossing vector, i.e., from positive to negative (z+) or from negative to positive (z−) and the corresponding *z*-current $I_{z}$. The zero-crossing vectors ${\vec{e}}_{\mathrm{zero},k}$ are indexed due to the detection time. Depending on this, the order of the cross product is swapped
(28)$${\vec{e}}_{\max}=\left\{\begin{array}{cc} \frac{{\vec{e}}_{\mathrm{zero},k}\;\times \;{\vec{e}}_{\mathrm{zero},k-1}}{\left\|{\vec{e}}_{\mathrm{zero},k}\;\times \;{\vec{e}}_{\mathrm{zero},k-1}\right\|}, & \mathrm{for}\;z+\;⋀\;\frac{dI_{z}}{dt}>0,\;\mathrm{or}\;z-\;⋀\;\frac{dI_{z}}{dt}<0\\ \frac{{\vec{e}}_{\mathrm{zero},k-1}\;\times \;{\vec{e}}_{\mathrm{zero},k}}{\left\|{\vec{e}}_{\mathrm{zero},k-1}\;\times \;{\vec{e}}_{\mathrm{zero},k}\right\|}, & \mathrm{else}. \end{array}\right.$$

### 3.3. Iterative Algorithm

For the proposed algorithm, knowledge of the geometry is required. This includes the relative position of the joint ${\vec{r}}_{j}$ and the distance of the sensor to that joint. The orientation of the sensor is aligned with the second bone. The corresponding setup is sketched in Figure 2. On the condition that the orientation of the sensor is correct, the proposed algorithm is convergent and delivers its position though applying the kinematic chain. Vice versa, the orientation determined for this position again matches the assumed position. To come to an iterative algorithm, we initially assume a random orientation ${\vec{\hat{e}}}_{s,i=0}$, and the kinematic chain model is used to determine a related estimate of the sensor position ${\vec{\hat{r}}}_{s,i}$: (29)$${\vec{\hat{r}}}_{s,i}={\vec{r}}_{j}+l_{\mathrm{js}}\;{\vec{\hat{e}}}_{s,i}.$$

From the procedure described in Equations (Equation 18)–(Equation 21) (Section 2.2), we next determine an update of the sensor orientation ${\vec{\hat{e}}}_{s,i+1}$ according to
(30)$${\vec{\hat{e}}}_{s,i+1}=\vec{f}\left({\vec{\hat{r}}}_{s,i},{\vec{e}}_{\max}\right).$$

This process is iteratively repeated until convergence is achieved. The number of iterations needed for this goal depends on the ratio between $l_{\mathrm{js}}$ and $|{\vec{r}}_{j}|$. The algorithm is represented as a flow chart in Figure 9. Figure 10 shows an example for the corresponding iterative progress. Based on the orientation of the previous chain element, we calculate the position of the next joint. Thus, the true alignment of the kinematic chain is found after a suitable number of repetitions of this systematic procedure.

The convergence speed of the algorithm depends on the ratio *Q* of the length from the actuator point to the joint $l_{\mathrm{aj}}$ and the length from this joint to the sensor $l_{\mathrm{js}}$:(31)$$Q=\frac{l_{\mathrm{js}}}{l_{\mathrm{aj}}}.$$

Figure 11 shows the amount of the absolute angular error as a function of the number of iterations for each *Q*. The true sensor orientation is assumed to be in the *z*-direction. The initial sensor orientation is always assumed to be in the *y*-direction, i.e., the corresponding angular error starts at $90^{\circ }$.

### 3.4. Uniqueness

In this section, we will show that the procedure described above delivers a unique result if the *Q* as defined in Equation (31) is **not** between 0.5 and 1. For a proof, we refer to Figure 5 and note that the orientation ${\vec{e}}_{s}$ has two components, i.e., two scalar degrees of freedom. The input of the algorithm has also two scalar given variables, which leads to a problem with two given variables and two unknowns. To solve this, we first split the two-dimensional solution vector and show that each can be calculated independently. As previously shown, ${\vec{e}}_{\max}$, ${\vec{e}}_{r}$, and ${\vec{e}}_{s}$ lie in one plane. From the sensor position, the joint position can be obtained by the vector addition of ${\vec{e}}_{s}$ scaled by the known length $l_{\mathrm{js}}$. This relationship shows that the joint position must also lie in the plane already shown. This allows the detected ${\vec{e}}_{\max}$ to be used to determine a plane in which the solution vector lies. This reduces the number of unknowns to 1 and the first degree of freedom can thus be determined unambiguously. In the previous derivation, the relative relationship between ${\vec{e}}_{r}$, ${\vec{e}}_{s}$, and ${\vec{e}}_{\max}$ was shown. In order to show that a unique ${\vec{e}}_{s}$ can be assigned to each ${\vec{e}}_{\max}$, a reference point must be selected. The angle ($\phi_{\max,j}$) is defined as between ${\vec{e}}_{\max}$ and the position vector of the joint ${\vec{r}}_{j}$ and the angle ($\phi_{s,j}$) is defined as being between ${\vec{e}}_{s}$ and ${\vec{r}}_{j}$. These angles are shown in Figure 12.

We set the coordinate system such that ${\vec{e}}_{r}$, ${\vec{e}}_{s}$ and ${\vec{e}}_{\max}$ lie in the xy-plane. Furthermore, without limiting the generalization of the solution we assume that ${\vec{r}}_{j}$ lies on the *y*-axis. The position of the joint and the sensor can thus be described as follows:(32)$${\vec{r}}_{j}=\left[\begin{array}{c} 0\\ l_{\mathrm{aj}} \end{array}\right]$$
(33)$${\vec{r}}_{s}=\left[\begin{array}{c} l_{\mathrm{js}}\cos\left(\phi_{s,j}\right)\\ l_{\mathrm{aj}}+l_{\mathrm{js}}\sin\left(\phi_{s,j}\right) \end{array}\right]$$

Now, the angle $\phi_{r}$ between ${\vec{r}}_{j}$, ${\vec{r}}_{s}$ can be determined as
(34)$$\phi_{r}=\arctan\left(\frac{l_{\mathrm{js}}\cos\left(\phi_{s,j}\right)}{l_{\mathrm{aj}}+l_{\mathrm{js}}\sin\left(\phi_{s,j}\right)}\right)$$

These descriptions can now be used to determine $\phi_{\max}$ according to Equation (Equation 17):(35)$$\phi_{\max}=-\arctan(\frac{1}{2}\tan\left(\phi_{s}-\phi_{\mathrm{rs}}\right)$$

This angle is now related to the position vector of the sensor. The $\phi_{\max,j}$ can be described as follows:(36)$$\phi_{\max,j}=\phi_{\max}-\phi_{r}$$

The obtained formulas reveal the analytical relationship $\phi_{\max,j}\left(\phi_{s,j}\right)$. For the uniqueness of the solution, the inverse function is required. Since this is not a trivial task, a lookup table has been created while the axes are swapped. Figure 13 shows the results obtained for different values of *Q*.

Obviously, there is a unique solution only for a certain range of *Q*. For Q=1, i.e., where the non-uniqueness is most significant, a simulation including a measurement of the convergence speed was carried out, see Figure 14. We observe that the maximum error does not approach zero, but oscillates periodically and is undamped around zero.

## 4. Results

### 4.1. Simulation Results

For validation, the proposed iterative algorithm was numerically implemented and the motion of a single joint was observed. To that end, an existing real-time framework based on C was applied [22]. An overview of the used modules is shown in Figure 15. A complete description of the implementation can be found in [23].

As an example, a magnetic source is attached at the beginning of a first bone of 40 cm length. The sensor is attached to the second bone at a distance of 10 cm to the joint. To not repeat the simulation without movement that has already been discussed in Section 3.3, a simple movement in the xz-plane was performed: The second bone rotates in the xz-plane around the *y*-axis in a range from $90^{\circ }$ to $-90^{\circ }$ with a constant angular frequency 1 Hz. As illustrated in Figure 16, the 3D coil (source) is localized in the origin at the end of the first bone with equivalent magnetic dipoles at frequencies $\omega_{\phi }=7\;\mathrm{Hz}$ and $\omega_{\theta }=7000\;\mathrm{Hz}$.

The results of the simulations are shown in Figure 17a. The ground-truth angle between the second element and the *z*-axis is shown for comparison with the estimated angle. For better visibility of both signals, the estimation is shifted by $10^{\circ }$. The absolute error of the amplitude which plotted in Figure 17b, is for most time steps in the range < 1°.

### 4.2. Experimental Results

For an initial laboratory test, a prototype consisting of two kinematic elements made of PVC was fabricated (see Figure 18). The two elements are connected by a screw, allowing one degree of freedom. A 3D coil is attached at the beginning of the longer element while a fluxgate magnetometer is attached to the shorter element. Firstly, tests were carried out with a static setup, i.e., the prototype was held in a fixed position using suitable wedges which was attached to the table and the prototype with double-sided adhesive tape.

We have used seven equidistant angles between 0° and 90° degrees. The same software used for the simulation has been applied for data acquisition and signal processing. A series of 10 s measurements was recorded for each position, and a new angle estimate was calculated every 20 ms. Hence, after 10 s we obtained 500 angle estimations represented in Figure 19.

### 4.3. Computational Cost

Determining the computational effort of an algorithm is not trivial when using libraries, as the implementation of the functions is not disclosed. Therefore, the mathematical operations used were given a score depending on their complexity. The score was determined by performing the corresponding operation one million times and normalizing it to the ADD operation. This should make it possible to compare the results with other algorithms. All operations are implemented in C with the “math.h” library.

Table 1 shows the calculation effort for the used mathematical operations. One iteration of the presented algorithm has an average equivalent of 212 addition operations. All operations were performed on an AMD Ryzen 5 5600X 6-Core processor.

## 5. Discussion

### 5.1. Convergence and Uniqueness

The convergence of the algorithm depends on the length ratio *Q*, i.e., the ratio of the distance between the actuator and the joint $l_{\mathrm{aj}}$ and the distance between the joint and the sensor location $l_{\mathrm{js}}$. It has been shown that the algorithm uniquely converges unless *Q* is between 0.5 and 1. Moreover, the number of iterations required for a given error threshold increases as *Q* increases. As illustrated in Figure 20, such behaviour sounds logical. For a fixed length of the first bone, the permissible angular range δ is limited by the length of the second one.

Regarding uniqueness, the role of *Q* may well lead to problems in a possible subsequent application, where care must be taken when designing and positioning the sensors and sources to ensure that the possible *Q* is not between 0.5 and 1.

### 5.2. Results

The algorithm was implemented in an existing real-time framework. A periodic movement was implemented using a simulation pipeline. The simulations have shown that for an unmoved kinematic chain, the estimate agrees with the simulated posture. Subsequently, a periodic movement with a frequency of 1 Hz was performed. The movement showed an error of approximately 1 degree. Outside the simulation, a larger deviation is to be expected. Due to the dipole approximation, small deviations occur in the modelling. In addition, noise was not used in the first investigations. This would not interfere with the algorithm but would worsen the result. Inaccuracies caused by noise may be improved or corrected by averaging the input signal. First experimental results indicate that the algorithm works in practice. It has been observed that the averaged values each have an offset to the true values. These deviations can be explained by the fact that there are model properties that have not yet been taken into account. For example, the algorithm assumes that the sensor is located exactly in the centre of the kinematic element. In this case, the sensor would move on a circular path. However, as the sensor cannot be located in the centre in reality, it tends to move along an elliptical curve. This leads to an error depending on the angle. The dipole approach leads to further errors. The closer the sensor is to the source, the worse the approximation becomes. In [25], it was shown that the deviation from the dipole approximation at a distance of 20 times the radius of the source is only 0.0027%. This is the case in our setup. If the algorithm is used in a setup where the distance cannot be kept large enough, approaches as in [26] can be used. The variance of the measurement series differs greatly. Particularly, it is unclear why the variance becomes smaller and smaller as the sensor angle increases. The signal quality was almost the same in all measurement series, so at first glance a similar variance could actually be assumed for each setup. However, the transfer function from the MV to the sensor projection is non-linear, which would explain a stronger fluctuation in the 0° range.

### 5.3. Computational Effort and Timings

For a motion capture system, the sample rate at which positions are captured is of particular importance. Therefore, a high sample rate is needed for fast movements. In human–machine interface applications, a low latency in the range of 15 ms is required. If the latency increases, surgery becomes more difficult for the user. To be able to guarantee such a maximum latency, the required computing time must be kept as low as possible. For this analysis, the implemented code was divided into each weighted individual operations. A computing time of 3 μs per joint and 60 μs for a kinematic construct consisting of a hand with up to 20 joints was determined. During the investigation, we have used a number of 15 iterations showing good results for the most setups. In a real application, this number could be even reduced. Because of the limited moving speed of each kinematic chain element, it is possible to use a well-fitting initial orientation. Especially for slow motions, this would reduce the number of needed iterations. Note that the investigation shows only a theoretically possible calculation time. In a real setup, there will be optimizations to the signal processing pipeline, primarily consisting of pre-processing and feature extraction. These additional parts will further increase the computational effort. Such promising results and ideas for finding optimal conditions lead to the outcome that the theoretically maximal latency can also be observed in a real application. Moreover, note that so far the implementation has been performed without any computational parallelization. As an example, the hand consists of several kinematic chains that can move independently of each other. A multi-threading implementation would thus be a further step to increase the efficiency and decrease latencies.

## 6. Conclusions and Outlook

This work has dealt with a new algorithm for motion tracking. To this end, we have introduced spatially rotating magnetic dipole sources. In this context, the maximum vector (MV) has been introduced as a new signal feature, and the spatial relationship between the MV, the sensor position, and its orientation were investigated. The correlations were linked to the model of a kinematic chain, such that this self-consistency was exploited and a computationally efficient algorithm was developed. The algorithm was implemented and validated in a real-time signal processing code. The performance was evaluated in terms of the accuracy of the results and the required computational effort. It was shown that the presented algorithm is very efficient in determining the posture of a kinematic chain. The theoretical functionality of the algorithm has already been demonstrated by simulation and with the realization of a first demonstrator. These were initially very simple movements with only one degree of freedom. More complex movements of real people are planned for a later stage of research. The behaviour of the algorithm in the presence of magnetic interference or in the event of a sensor failure has not yet been investigated and will be addressed in the future.

The algorithm currently uses a strong simplification of a kinematic chain. In addition, the modelling must take into account further parameters such as the thickness, the precise position of the sensor on the element and a potential tilting of the sensor. A consideration of more details would increase the number of model parameters, and an automatic/semi-automatic calibration procedure should be developed to keep it manageable. Moreover, an additional automated distance measure could be beneficially for the stability of the outcomes. Finally, as a 3D coil generates a signal that provides three independent pieces of information and in the current implementation only two phases of information have been used, from the absolute signal value, more distance information can be obtained. This could be integrated with a Kalman filter approach as in [27] and/or correct a parameter of the model during runtime so that the error can be minimized.

## 7. Patents

The content of this paper was used for a patent application. It was granted by the German Patent Office with the patent number DE 10 2023 119 167.

## Figures and Tables

**Figure 1 sensors-24-06947-f001:**
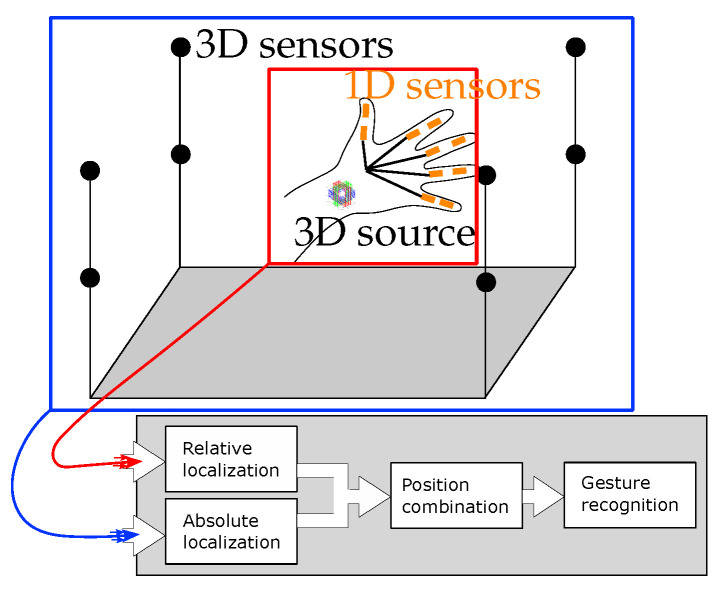
System overview: The illustration includes an external localization (blue) consisting of a defined setup of (here 8) sensors. The inner localization (red) consists of a 3D coil which is attached to the wrist as well as magnetic 1D sensors which are attached to each finger element. Following the localization, gesture recognition or processing of the data for the human–machine interface can be carried out.

**Figure 2 sensors-24-06947-f002:**
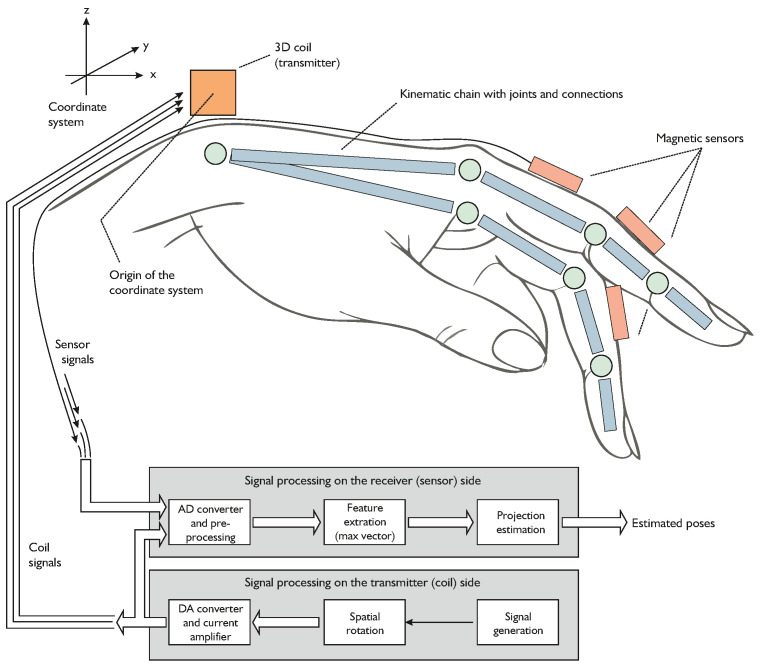
Typical example of use of the presented algorithm: At the origin of the coordinate system a 3D magnetic transmitter is located. A kinematic chain is equipped with 1D magnetic sensors, such as fluxgate magnetometers or magnetoelectric sensors, on every chain element. The kinematic chains are connected through joints with ellipsoidal cross-sections, each providing two degrees of freedom. Any additional information from the kinematic chain about the position is used to increase the speed of the localization algorithm.

**Figure 3 sensors-24-06947-f003:**
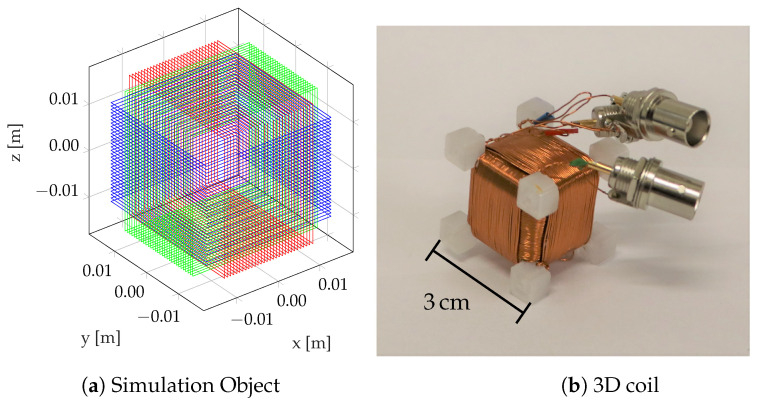
3D coil: (**a**) sketches the modelled simulation object. A photograph of the corresponding realization is shown in (**b**). Note that both constructions consist of three orthogonal coils.

**Figure 4 sensors-24-06947-f004:**
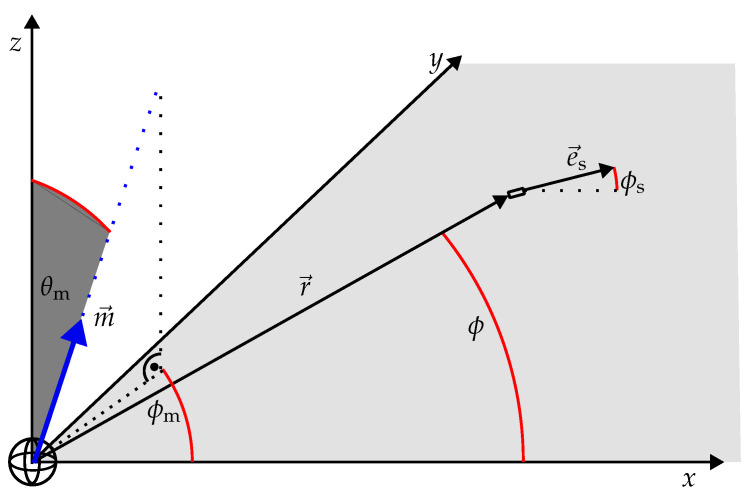
Geometry used for the derivation: $\vec{r}$ and ${\vec{e}}_{s}$ both lie in the xy-plane. $\phi_{m}$ and $\theta_{m}$ define the orientation of the rotating magnetic dipole $\vec{m}$ in spherical coordinates.

**Figure 5 sensors-24-06947-f005:**
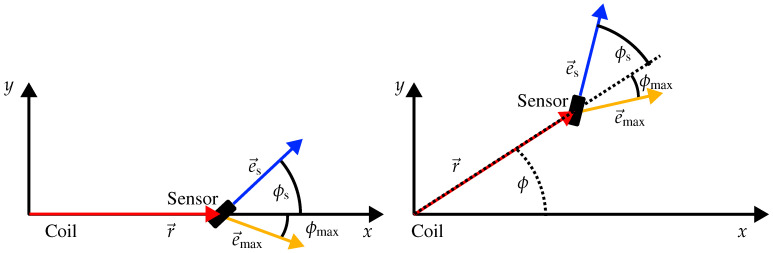
The relation in Equation (Equation 17) is independent of the angle ϕ. Moreover, the unique relationship between the three unit vectors ${\vec{e}}_{s},{\vec{e}}_{max},$ and ${\vec{e}}_{r}$ is clarified.

**Figure 6 sensors-24-06947-f006:**
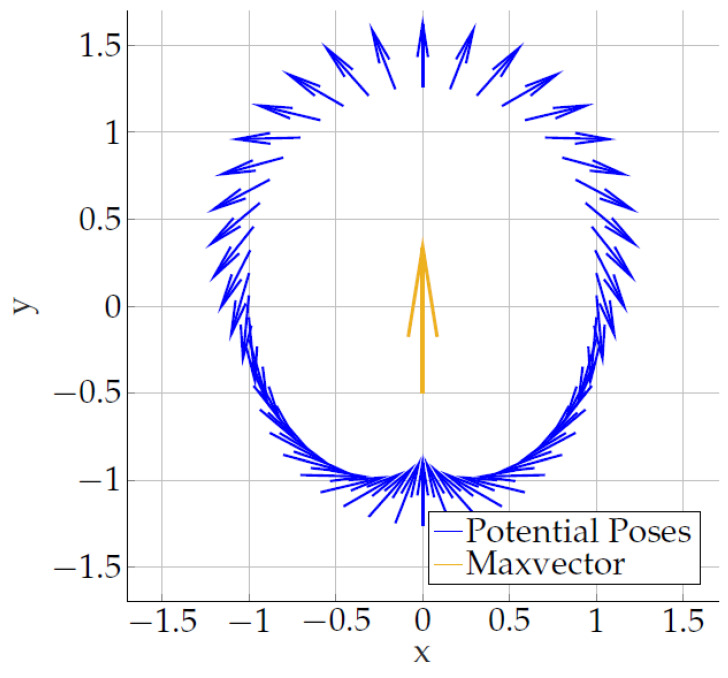
Blue vectors: Calculated sensor orientations ${\vec{e}}_{s}$ for different values of the sensor location $\vec{r}$. The starting point of each blue vector represents the corresponding $\vec{r}$. Yellow vector: The maximum vector at the origin, always polarized in the *y*-direction. Note that the lengths of the blue vectors are not of interest here, as only the directions are relevant.

**Figure 7 sensors-24-06947-f007:**
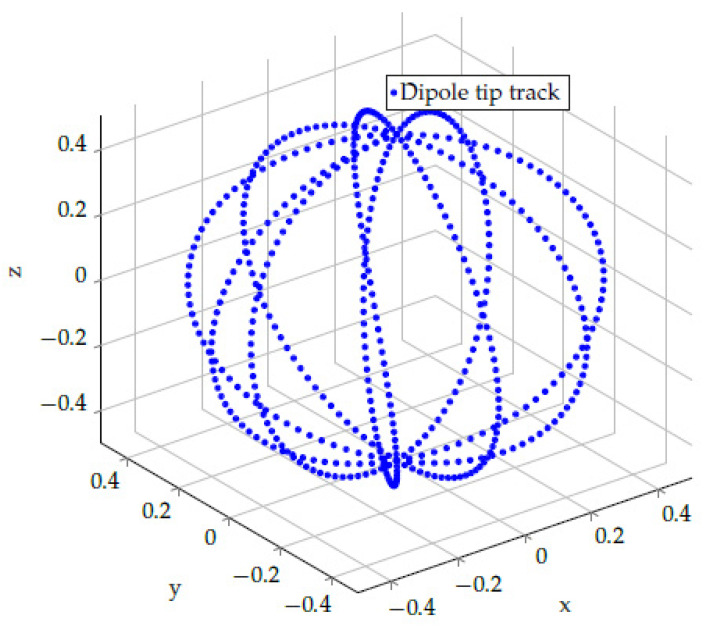
Track of the magnetic dipole $\vec{m}\left(t\right)$ with starting point at the origin as a function of time. The tip of $\vec{m}$ moves on the surface a sphere with radius $m_{0}$, according to Equation (Equation 22) for $N_{\omega }=\omega_{\theta }/\omega_{\phi }=10$.

**Figure 8 sensors-24-06947-f008:**
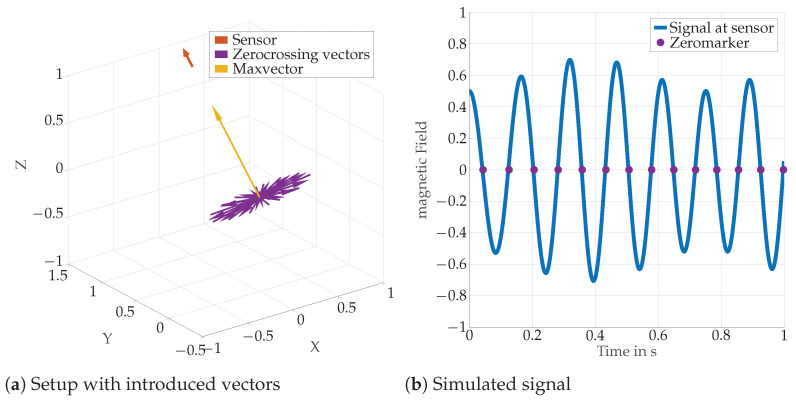
The left figure exemplary shows a max vector at the origin (yellow), the position and orientation (orange) of the sensor, and the corresponding plane of zero-crossing vectors (purple). The right side shows the corresponding sensor signal as a function of time. The times when a zero-crossing is achieved are marked with a purple dot. The simulation works with a source which is driven with $N_{\omega }=\omega_{\theta }/\omega \phi =10$. The absolute values/lengths are not relevant, as the relative relationship between the vectors and the zero crossings are both of interest.

**Figure 9 sensors-24-06947-f009:**
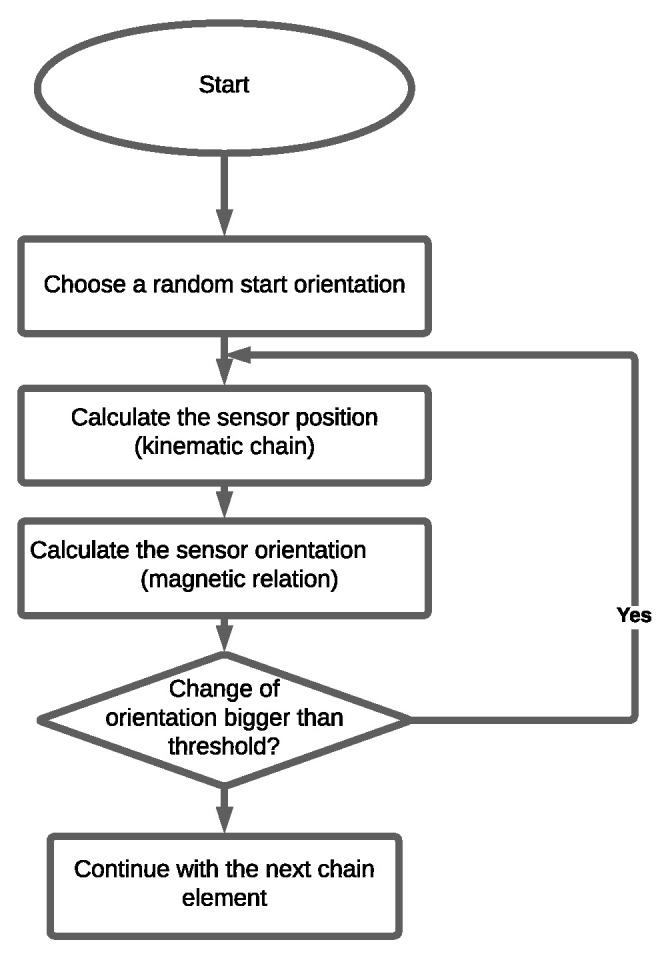
Flow chart of the iterative algorithm: The algorithm starts with a random initial orientation. Then, the sensor position relative to the source is determined. Afterwards, the corresponding orientation is calculated. When there is no relevant change between the data obtained with two subsequent iterations, convergence is reached, and this orientation is the estimated result.

**Figure 10 sensors-24-06947-f010:**
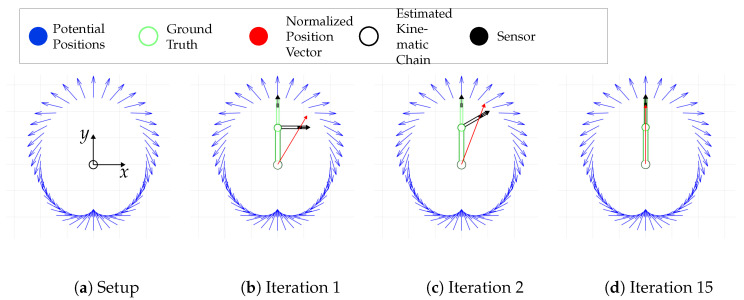
Exemplary iterative process: This figure shows the iterations for a simple setup. The first sub-figure shows the used setup with the coordinate system. The origin of this setup is located at the first kinematic chain element, where the source is also located. The source is attached to the first kinematic element in such a way that the relative position of the source to the kinematic chain is always constant. In the following figures, the coordinate system has been omitted. The blue vectors represent the potential poses for the detected MV. The light green construction shows the ground truth. The red vector is a normalized position vector of the sensor which points to the potential pose in this direction. The sensor is mounted on the second bone. It is represented by a black rectangle with a vector in the sensitive direction. Subfigure (**b**) starts with a bone orientation in *x*-direction. For some iterations, the kinematic chain and the related position vector are shown. After 15 iterations, subfigure (**d**), the sensor pose matches a potential pose (ground truth) and the algorithm converges.

**Figure 11 sensors-24-06947-f011:**
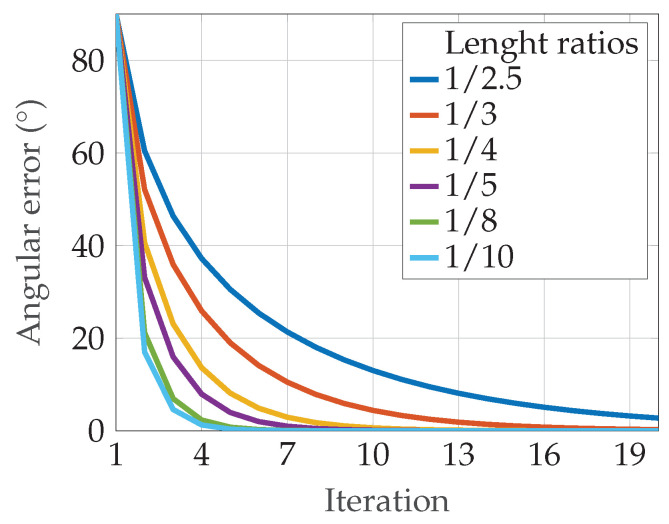
Angular error in dependence of the iteration: The figure shows the behaviour of the angular error in dependence of the number of iteration. Different setups of length ratios are looked at. The legend shows the corresponding *Q* for each curve. All curves tend closer to zero with each iteration.

**Figure 12 sensors-24-06947-f012:**
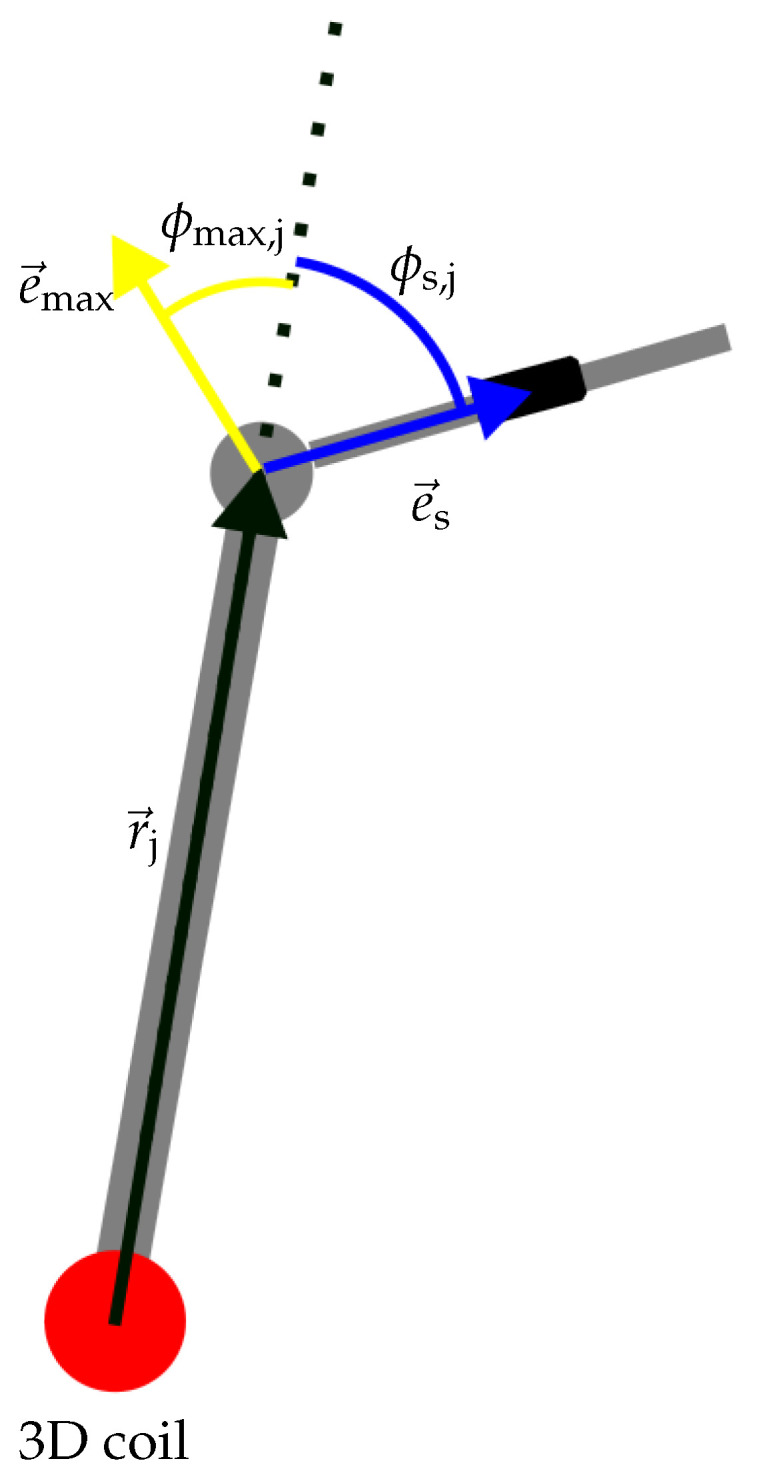
Definition of the angles at a joint between two bones.

**Figure 13 sensors-24-06947-f013:**
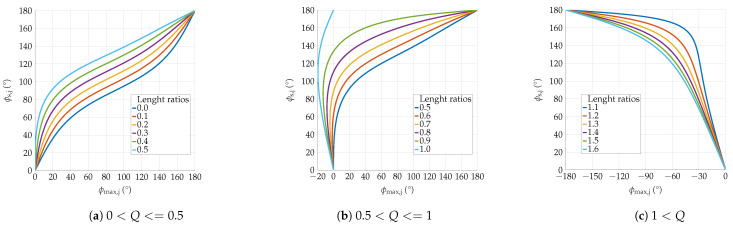
The relation between $\phi_{s,j}$ and $\phi_{\max,j}$ is represented for different values of Q. The plots are subdivided for different values of *Q*. In the ranges 0<Q<0.5 and Q>1 there is a clear assignment, i.e., there is a unique bidirectional relation between $\phi_{\max,j}$ and $\phi_{s,j}$. However, between 0.5 and 1 we observe a non-unique relation.

**Figure 14 sensors-24-06947-f014:**
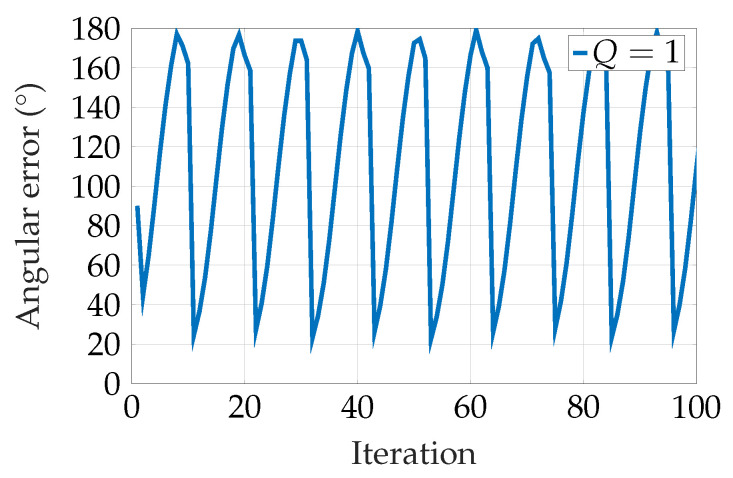
For Q=1, the maximum angular error does not approach zero even after several iterations, i.e., the algorithm is non-convergent.

**Figure 15 sensors-24-06947-f015:**
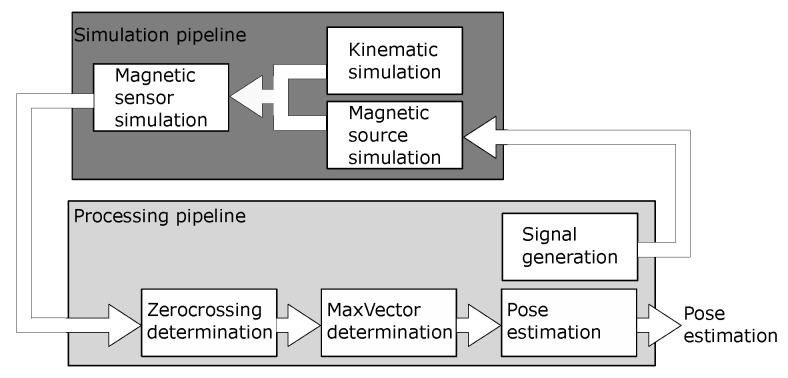
Simulation overview: The simulation is divided in two sections. The upper (dark) part simulates the motion and the resulting field at the sensor. In the lower part (bright), the described algorithm is implemented and the pose is calculated.

**Figure 16 sensors-24-06947-f016:**
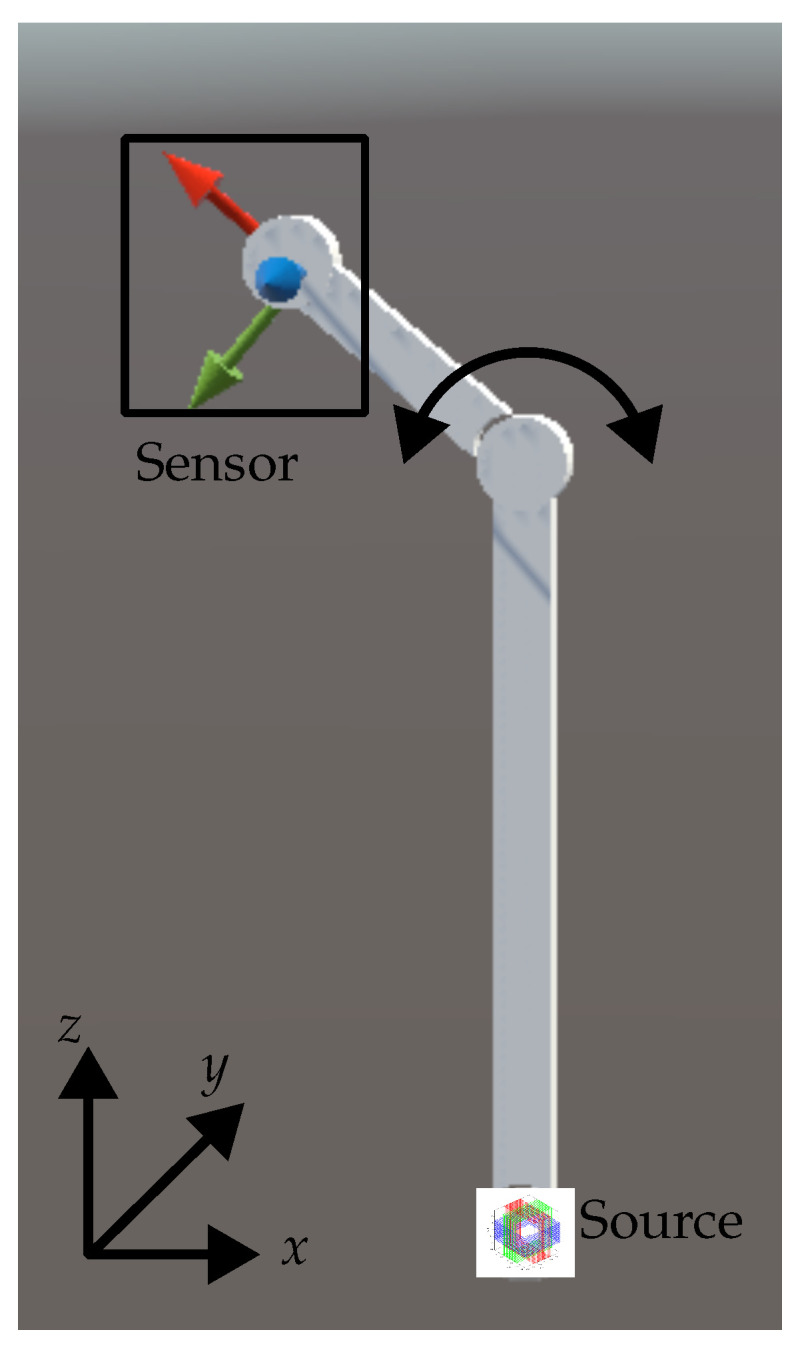
Simulation of a motion: All elements are in the yz-plane. The 3D coil source is located in the origin. The first bone is aligned with the *z*-axis and its end represents the position of the joint. The second bone moves from $90^{\circ }$ to $-90^{\circ }$ with respect to the axis of the first bone.

**Figure 17 sensors-24-06947-f017:**
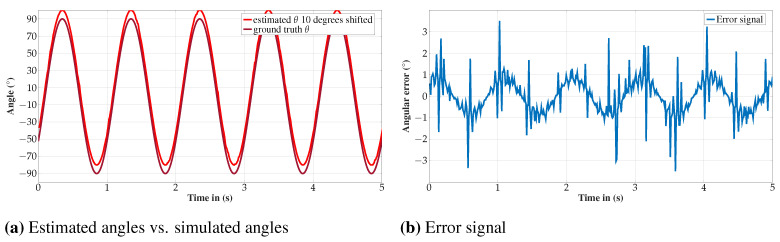
(**a**) shows both the estimated angle and the simulated one. The dark red line represents the simulation (ground truth) while the light red line is the estimation of the described algorithm. The latter is shifted 10° to enhance the clarity of the visualization. In (**b**), the difference between the simulation and the estimation is plotted. We observe an error signal which follows the angle of the movement.

**Figure 18 sensors-24-06947-f018:**
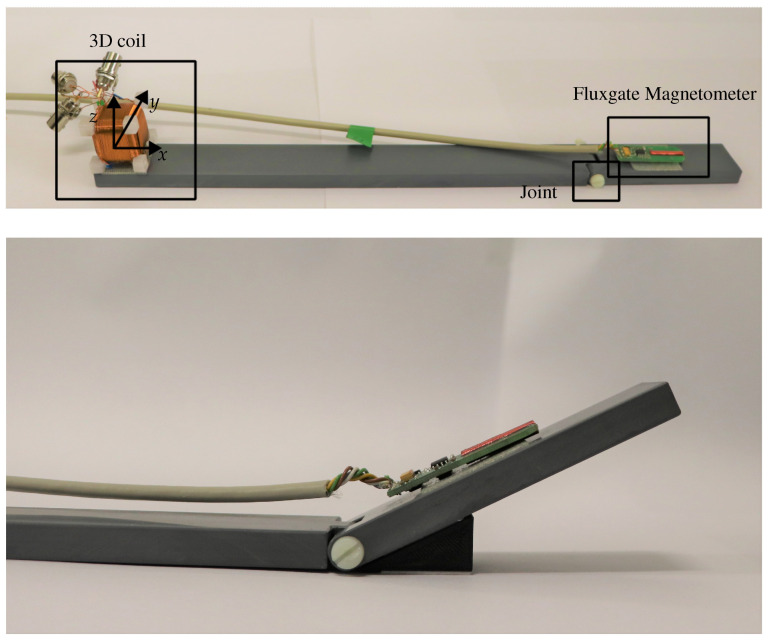
Upper figure: The built prototype consists of two PVC elements, connected to each other with a screw allowing for one degree of freedom. The 3D coil source is located at one end of the longer element. On the shorter element, a fluxgate magnetometer [24] is mounted. The illustration in the lower figure shows the assembly for a 30° position.

**Figure 19 sensors-24-06947-f019:**
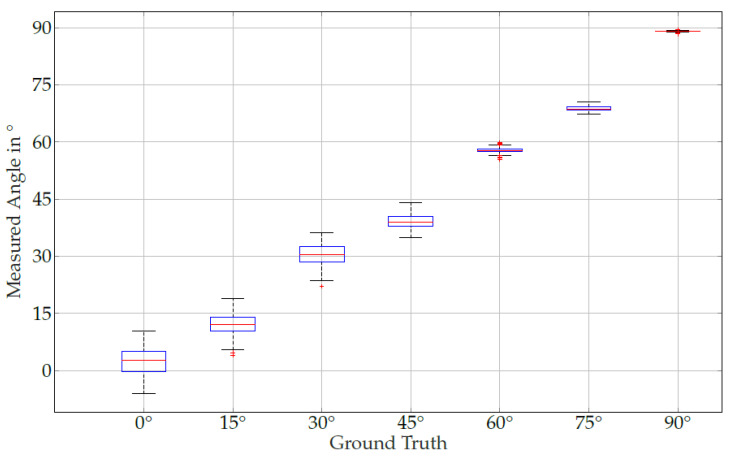
The boxplots show the experimental results of the measured sensor angles for each of the seven given (ground-truth) joint angles. The box plots show the median, the first quartile, the third quartile, the minimum, the maximum, and several outliers for each joint angle.

**Figure 20 sensors-24-06947-f020:**
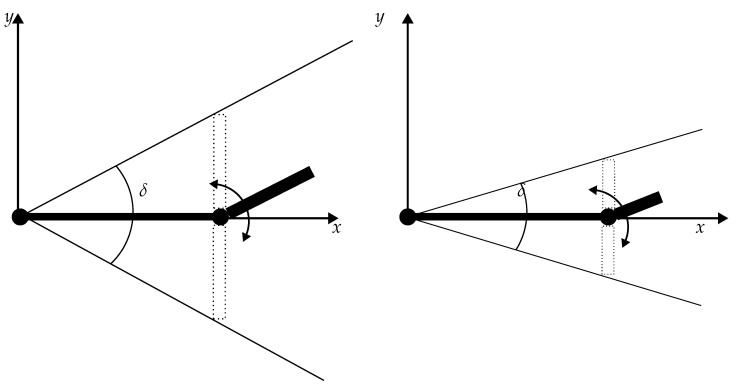
Possible angle ranges δ for two different lengths of the second bone at a fixed length of the first bone (*Q* is higher for the left realization).

**Table 1 sensors-24-06947-t001:** Time consumption of different complex operations for one million executions, normalized to the (basic) addition operation.

Operation	Time Consumption/μs	Score
ADD/SUB	940	1
MULT/DIV	960	1.02
SIN/COS/TAN	4970	5.29
ARCCOS/ARCSIN/ARCTAN	68,000	72.34
SQUARE ROOT	3250	3.46
CROSSPRODUCT	3700	3.94
SCALARPRODUCT	2200	2.34
Algorithm1Iteration	199,280	212.08

## Data Availability

The data presented in this study are available on request from the corresponding author.

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
