# Peer review of "A New Iterative Algorithm for Magnetic Motion Tracking"

_sensors, 2024, doi:10.3390/s24216947_

Round 1
Reviewer 1 Report
Comments and Suggestions for Authors
This paper introduces an innovative magnetic motion tracking algorithm for hand motion tracking applications. The paper demonstrates the novelty of the theory, and implements the algorithm in a real-time framework and carries out preliminary validation. The structure of the paper is clear, and the logical flow from problem statement to algorithm design, implementation, evaluation and discussion is coherent and easy to understand. In addition, the paper also considers the computational costs in practical applications, which are essential for actual deployment. This paper presents an iterative algorithm based on "maximum vector" feature, which provides a new method for magnetic motion tracking. The algorithm has low computational complexity, and its effectiveness and accuracy have been verified by simulation and laboratory tests. The structure of the paper is clear, the logic is rigorous, easy to understand. However, this paper also has some problems: the convergence of the algorithm with respect to the length ratio Q is analyzed, but other influencing factors such as sensor noise and magnetic field distortion are not discussed. The innovation of the algorithm mainly lies in the improvement of the existing theory, but it fails to solve the uniqueness problem in all cases, especially in some Q value range (between 0.5 and 1), the algorithm shows non-uniqueness. The modeling of the motion chain is too simple and does not take into account the thickness of the motion chain and the position of the sensor, which may affect the accuracy of the algorithm. There is no analysis of the robustness of the algorithm, such as its performance under magnetic interference or sensor failure. The experimental data source is limited, and only a few tests have been carried out, which cannot fully prove the applicability of the algorithm in complex motion scenes. Validation is limited to simple PVC Settings and lacks experimental data in real-world scenarios, such as human joint movements.
Comments on the Quality of English LanguageThe paper as a whole is well written, the use of professional terms is appropriate, and the algorithm steps are clearly described. The system design and experimental procedure are described succinctly.
There are some minor grammatical problems, such as occasional misphrasing (for example, "calculate the corresponding direction", which can be rephrased as "then calculate the direction").
In order to improve readability, the reuse of certain terms, such as "chain of motion", can be reduced.
Author Response
Dear reviewers,
we would like to thank you for your detailed comments and suggestions. We have revised our
manuscript and are convinced that your comments have improved the quality of our work.
Please find attached the revised manuscript. The corresponding changes and refinements
made in the revised manuscript are shortly summarized in the responses below. A PDF with the marked changes is attached.
Yours sincerely,
Tobias Schmidt
Comment 1: The modeling of the motion chain is too simple and does not take into account the thickness of the motion chain and the position of the sensor, which may affect the accuracy of the algorithm.
Response 1: Thank you for this thought. The comment is correct and a more precise description of the kinematic chain would be an improvement in the accuracy aspect. This aspect will be considered in the further research of the algorithm. In this paper, we have therefore adjusted the outlook and put more focus on making it clear that this could still be a problem.
Therefore we changed in the last paragraph to stress the point that here is some more work to do. “The algorithm currently uses a strong simplification of a kinematic chain. In addition, the modeling must take into account further parameters such as the thickness, the precise position of the sensor on the element and a potential tilting of the sensor.”
Comment 2: There is no analysis of the robustness of the algorithm, such as its performance under magnetic interference or sensor failure.
Response 2: The comment is correct. The robustness of the algorithm against magnetic interference or sensor failure has not been investigated in this paper and will play a role in the further course. We have added a sentence to clarify this. We added one sentences in the conclusion and outlook part:
“The behavior of the algorithm in the presence of magnetic interference or in the event of a sensor failure has not yet been investigated and will be addressed in the future.”
Comment 3: The experimental data source is limited, and only a few tests have been carried out, which cannot fully prove the applicability of the algorithm in complex motion scenes. Validation is limited to simple PVC Settings and lacks experimental data in real-world scenarios, such as human joint movements.
Response 3: This comment is correct. In this work, only a concept and an initial proof of concept were shown. Further and more complex movements will be carried out in the future. We have added a few sentences in the conclusion and outlook.
“These were initially very simple movements with only one degree of freedom. More complex movements of real people are planned for a later stage of research.”

Reviewer 2 Report
Comments and Suggestions for Authors
In this manuscript, the authors propose an algorithm for pose estimation of a kinematic chain based on magnetic localization principles. In specific, they analytically identify the direction for which the maximum magnetic field magnitude at the location of the field sensors is received from a rotating magnetic dipole source (3D coils), linking the location vector (between source and sensor) and the sensor orientation. Then, they use this analysis to develop an iterative algorithm with low complexity that identifies the posture of the kinematic chain, taking also into account the structure of the kinematic chain. The authors verify the efficiency of the proposed method in simulations, achieving errors in the range of 1° and execution times of 3 μs per joint, and also validate its performance through laboratory tests. The paper is in general interesting and well-written. Some minor comments:
· I would propose to enhance the introduction with 1-2 paragraphs that clarify the innovative aspects of your work.
· There are some minor typos, e.g., line 136, in Figure 4 not all the unit vectors are shown, the equation before line 186 is unnumbered. Please carefully proofread your manuscript.
· Although you have identified the uncertainty introduced in your method, I think that it would be beneficial to include some references related to the dipole approximation, as well as the accuracy of the distance between source and sensor. Relevant references:
1. "Magnetic dipole model in the near-field." 2015 IEEE International Conference on Information and Automation. IEEE, 2015.
2. "Automated Estimation of Magnetic Sensor Position and Orientation in Measurement Facilities." IEEE Transactions on Electromagnetic Compatibility (2024).
3. "Measurement of the magnetic signature of a moving surface vessel with multiple magnetometer-equipped AUVs." Ocean Engineering 64 (2013): 80-87.
4. "A software-based calibration technique for characterizing the magnetic signature of EUTs in measuring facilities." IEEE Transactions on Electromagnetic Compatibility 59.2 (2016): 334-341.
5. "Magnetic field analysis for distance measurement in 3D positioning applications." 2016 IEEE International Instrumentation and Measurement Technology Conference Proceedings. IEEE, 2016.
· Moreover, some of the above references present methods that are based on stochastic techniques (e.g., genetic algorithms) or even machine learning. Please elaborate on whether it would be beneficial for your method to integrate various techniques (e.g., ML-assisted pose estimation).
Author Response
Dear reviewers,
we would like to thank you for your detailed comments and suggestions. We have revised our
manuscript and are convinced that your comments have improved the quality of our work. The corresponding changes and refinements made in the revised manuscript are shortly summarized in the responses below. Please find attached a PDF of the work with the changes in red.
Yours sincerely,
Tobias Schmidt
Comment 1: I would propose to enhance the introduction with 1-2 paragraphs that clarify the innovative aspects of your work.
Response 1: Thank you for the comment. We have added a paragraph in the introduction to show the advantages and innovative aspects more clearly.
“The advantage of the presented algorithm is that it combines localization and mapping to a kinematic chain. In this way, prior knowledge about a kinematic chain is integrated into a localization, thus narrowing down the solution space and simplifying the calculation.”
Comment 2: There are some minor typos, e.g., line 136, in Figure 4 not all the unit vectors are shown, the equation before line 186 is unnumbered. Please carefully proofread your manuscript.
Response 2: Thank you for the comment. We have corrected your comments and also proofread and corrected other minor errors.
Comment 3: Although you have identified the uncertainty introduced in your method, I think that it would be beneficial to include some references related to the dipole approximation, as well as the accuracy of the distance between source and sensor. Relevant references:
- "Magnetic dipole model in the near-field." 2015 IEEE International Conference on Information and Automation. IEEE, 2015.
- "Automated Estimation of Magnetic Sensor Position and Orientation in Measurement Facilities." IEEE Transactions on Electromagnetic Compatibility(2024).
- "Measurement of the magnetic signature of a moving surface vessel with multiple magnetometer-equipped AUVs." Ocean Engineering64 (2013): 80-87.
- "A software-based calibration technique for characterizing the magnetic signature of EUTs in measuring facilities." IEEE Transactions on Electromagnetic Compatibility59.2 (2016): 334-341.
- "Magnetic field analysis for distance measurement in 3D positioning applications." 2016 IEEE International Instrumentation and Measurement Technology Conference Proceedings. IEEE, 2016.
Response 3: Thank you for this comment. The deviation caused by the dipole approximation should not be neglected. We found her first source suggestion suitable for our work. Furthermore, we added an additional source that analysed the deviation of a cylindrical coil with the dipole approximation. “Cylindrical induction coil to accurately imitate the ideal magnetic dipole“, Eugene Paperno, Anton Plotkin
“The dipole approach leads to further errors. The closer the sensor is to the source, the worse the approximation becomes. In [25] it was shown that the deviation from the dipole approximation at a distance of 20 times the radius of the source is only 0.0027\%. This is the case in our setup. If the algorithm is used in a setup where the distance cannot be kept large enough, approaches as in [26] can be used.
”
Comment 4: Moreover, some of the above references present methods that are based on stochastic techniques (e.g., genetic algorithms) or even machine learning. Please elaborate on whether it would be beneficial for your method to integrate various techniques (e.g., ML-assisted pose estimation).
Response 4: Thank you for this suggestion. We will take this into account in the further course of our research.
